# Optimizing Treatment Strategies for *Egfr*-Mutated Non-Small-Cell Lung Cancer Treated with Osimertinib: Real-World Outcomes and Insights

**DOI:** 10.3390/cancers16213563

**Published:** 2024-10-23

**Authors:** Quentin Dominique Thomas, Nicolas Girard, Lise Bosquet, Sarah Cavaillon, Thomas Filleron, Siham Eltaief, Christos Chouaid, Hervé Lena, Didier Debieuvre, Maurice Perol, Xavier Quantin

**Affiliations:** 1Department of Medical Oncology, Institut du Cancer de Montpellier, 34000 Montpellier, France; sarah.cavaillon@icm.unicancer.fr (S.C.); xavier.quantin@icm.unicancer.fr (X.Q.); 2Department of Medical Oncology, Institut Curie, 75005 Paris, France; nicolas.girard2@curie.fr; 3Unicancer, 75000 Paris, France; l-bosquet@unicancer.fr (L.B.); s-eltaief@unicancer.fr (S.E.); 4Biostatistics Unit, Oncopôle Claudius Régaud IUCT-O, 31100 Toulouse, France; filleron.thomas@iuct-oncopole.fr; 5Department of Pneumology, Centre Hospitalier Intercommunal, 94010 Créteil, France; christos.chouaid@chicreteil.fr; 6Pneumology Department, CHU Pontchaillou, 35000 Rennes, France; herve.lena@chu-rennes.fr; 7Respiratory Medicine Department, Groupe Hospitalier de la Région Mulhouse Sud-Alsace, 68100 Mulhouse, France; debieuvred@ghrmsa.fr; 8Department of Medical Oncology Centre Léon Bérard, 69008 Lyon, France; maurice.perol@lyon.unicancer.fr

**Keywords:** non-small-cell lung cancer, tyrosine kinase inhibitors, *EGFR* mutation, treatment sequencing, brain metastasis

## Abstract

This study examined the real-world effectiveness of osimertinib, a third-generation tyrosine kinase inhibitor, on EGFR-mutated non-small-cell lung cancer. Osimertinib has shown promising results in clinical trials, but this research study looked at how it performs in real-world settings, where patients may have different characteristics than those typically included in trials. This study reviewed data from over 600 patients treated with osimertinib, either as their first (L1) or second metastatic line (L2) of treatment. The results showed that patients who took osimertinib as L1 experienced a median progression-free survival of 12.4 months, compared to 7.4 months for those treated in L2. The presence of brain metastases was associated with a poorer prognosis. Overall survival rates were lower than those reported in clinical trials, likely due to differences in patient conditions, such as older age and worse overall health. This study highlights the challenges of applying trial results to everyday medical practice and emphasizes the need for tailored treatment approaches for patients with advanced lung cancer.

## 1. Introduction

Lung cancer is the leading cause of cancer-related deaths worldwide [1]. Non-small-cell lung cancer (NSCLC) is the most common histological type, accounting for 80–85% of cases, and is often diagnosed at an advanced or metastatic stage [2]. In this context, comprehensive molecular genotyping aims to identify oncogene-addicted tumors [3]. One of the most frequent alterations is mutations in the epidermal growth factor receptor (*EGFR*), which confer sensitivity to *EGFR* tyrosine kinase inhibitors (TKIs). This represents a landmark application of precision medicine for NSCLC [4]. In over 85% of cases, these mutations are in-frame deletions in exon 19 or a single amino acid mutation, *L858R*, in exon 21 [5].

Osimertinib, a third-generation *EGFR* TKI, has been developed specifically to target the *EGFR T790M* mutation, which is associated with acquired resistance to first- and second-generation EGFR TKIs [6]. Osimertinib subsequently demonstrated progression-free survival (PFS) and overall survival (OS) benefits compared to first-generation *EGFR* TKIs in the first-line setting in the FLAURA trial, with a median OS of 38.6 months (95% CI [34.5 to 41.8]) vs. 31.8 months (95% CI [26.6 to 36.0]; *p* = 0.046), establishing the current standard of care upfront, particularly in the final FLAURA results [7].

Despite the central nervous system (CNS) efficacy observed with osimertinib, outcomes for patients with baseline CNS metastases remain poorer than for those without [8]. Ultimately, the application of therapeutic strategies from clinical trials to real-world practice remains uncertain, as outcomes may vary due to the significant impact of subsequent therapies and baseline clinical characteristics, which might not be accurately reflected in clinical trials [9]. Here, we utilized a large, nation-wide cohort of consecutive patients with advanced, metastatic *EGFR*-mutant NSCLC to report the therapeutic strategies and outcomes of patients treated in a real-world setting, with a focus on CNS metastases. This large database has already been used to describe clinical characteristics and survival outcomes of patients with advanced NSCLC with oncogenic alterations [10,11].

## 2. Material and Methods

### 2.1. Study Design

This non-interventional, retrospective, comparative study aimed to describe the treatment strategies and outcomes of *EGFR*-mutant, advanced or metastatic NSCLC patients included in the Epidemiological Strategy and Medical Economics (ESME) lung cancer (LC) data platform [12]. The ESME LC database (ClinicalTrial.gov NCT03848052) is a centralized, de-identified, structured database derived from the electronic health records of consecutive patients treated for LC since 2015 at one of the 38 participating health facilities (18 private non-profit Comprehensive Cancer Centers spread over 20 sites and 18 University or General Hospitals). Patient-related data, hospitalization-related data, and pharmacy-related data are collected, including patient demographic characteristics, pathology, and disease evolution. Treatment strategies are also recorded. Data are updated yearly and undergo a data management process aimed at ensuring the quality of the data analyzed.

Unicancer manages the ESME LC data platform in accordance with current best practice guidelines. In compliance with French regulations, the ESME Data Warehouse was authorized by the French data protection authority (initial authorization no. DE-2017-397 and subsequent amendment dated 14 October 2019, in accordance with GDPR). All patients approved the use of their data.

### 2.2. Patient Population

In this study, patient selection focused on patients with the following inclusion criteria:
-Histologically confirmed advanced NSCLC;-Common *EGFR*-activating mutations (i.e., exon 19 deletion or *L858R* mutation in exon 21);-Patients treated with osimertinib in the first (L1) or second (L2, i.e., administered after first- or second-generation TKIs) line of treatment for advanced disease;-Patients either newly diagnosed with or relapsed from early-stage disease.

Patients with one of the following exclusion criteria were excluded from this study:
-Uncommon *EGFR*-activating mutations identified;-Patients treated with another regimen than *EGFR* TKIs in the first metastatic line;-Patients who received first- or second-generation *EGFR* TKIs as first-line treatment for metastatic disease were excluded if they had not been treated with osimertinib monotherapy as second-line therapy.

Data analyzed were those available in the ESME LC database as of the data cut-off of 1 October 2022. In France, it is recommended that CNS imaging be performed at baseline for all patients, irrespective of neurological symptoms.

### 2.3. Data Collection

The following data were collected: sociodemographic characteristics (gender, age, weight, height, body mass index, smoking status); clinical characteristics (medical history, history of other cancers, performance status); disease characteristics (date of primary tumor diagnosis, stage at primary tumor diagnosis, histological subtype of NSCLC, setting of advanced NSCLC (i.e., de novo or progression of regional disease or metastatic disease), time from primary tumor diagnosis to advanced NSCLC diagnosis, location of metastases, time from advanced NSCLC diagnosis to treatment initiation); *EGFR* mutation and other mutations (e.g., *ALK*, *KRAS*, *BRAF*, *ROS1*); treatment strategy (therapeutic agents, start and stop dates); disease evolution over time; survival status. Treatment lines were derived a posteriori based on treatment dates and progression events: a treatment line was defined as a given therapeutic strategy delivered for metastatic NSCLC until progression. The start date of the first-line therapy was defined as the date of the first systemic treatment initiated after diagnosis. Disease progression was defined as the occurrence of a new metastatic site or progression of existing metastases, regional tumor or lymph node progression, or discontinuation of systemic treatment due to progression or death from any cause. For the purpose of this study, the index date was defined as the start date of osimertinib.

### 2.4. Statistical Analyses

Data were summarized as medians and ranges for continuous variables and as frequencies and percentages for categorical variables. The number of missing data was presented for each variable but not considered in the percentage calculations. The Kaplan–Meier method was used to estimate survival endpoints (OS and PFS). OS was defined as the time from the locally advanced or metastatic diagnosis date to the date of death from any cause. OS(id) was defined as the time from the index date to the date of death from any cause. PFS was defined as the time from the start date of the line (=index date) to the date of the first progression of the disease or death from any cause; patients without progression reported after the index date or with death reported more than 9 months after the date of the last medical information were censored at the date of the last medical information. Time to CNS metastases was defined as the time from the index date to the onset of CNS metastases in the osimertinib line. Patients with no onset of CNS metastases recorded were censored at the date of the last medical information or subsequent line, whichever occurred first. The HR and associated 95% CI were calculated using a Cox proportional-hazards model. Multivariable Cox proportional hazards models were constructed using a backward step-by-step manual selection procedure to identify time-independent prognostic factors of OS(id) in the whole cohort and for each treatment group. All factors significant at a conservative 2.5% level in univariable analysis were included in multivariable analysis. The variables included in the model were as follows: performance status; metastatic site (brain, liver, bone); age; gender; smoking status; *EGFR* mutation type; *T790M* mutation in exon 20; PD-L1 status; body mass index; radiotherapy at index date. The final model was reached when including only significant factors at a *p* < 0.05 significance level. Comparisons between survival at fixed points in time were conducted using the Cloglog method according to Klein [13]. Statistical analyses were performed using SAS version 9.4 (SAS Institute, Inc., Cary, NC, USA) and R Core Team (2016, version 4.1.2).

## 3. Results

### 3.1. Characteristics of EGFR-Mutated Patients Treated with Osimertinib for Metastatic NSCLC

Of the 39,974 patients in the ESME LC database at the time of analysis, with a median follow-up of 49.7 months (95% CI [49.0–50.6 months]), 29,600 had confirmed advanced NSCLC. Overall, 16,759 NSCLC patients had *EGFR* testing at least once, 2569 of whom were *EGFR*-mutant (EGFRm) NSCLC (15.3%). Among these patients, 1764 presented with common *EGFR* mutations (deletion in exon 19 or *L858R* mutation in exon 21). Of these, 1262 patients were treated with first- or second-generation *EGFR* TKIs in the first line (L1), and 426 patients (33.8%) received osimertinib as monotherapy in the second line (L2). Finally, 624 patients received osimertinib as monotherapy in either L1 (198 patients) or L2 (426 patients) and were included in this study (Figure 1).

The majority of the study population were female (73.4%; n = 458); the median age was 70.0 years (range: 28–93). Current or former smokers (regardless of the number of pack-years, i.e., ≥1) accounted for 41.1% of the population, and 16.2% had a history of another cancer. The most common metastatic sites at the index date were bone (61.6%; n = 372), brain (46.7%; n = 282), including symptomatic brain metastasis (70.9%; n = 200/282), and contralateral lung (35.9%; n = 217). Notably, among patients treated with osimertinib in L1, 83 (44.6%) had brain metastasis, including 52/83 with symptomatic brain metastasis (62.7%). Among those who received osimertinib in L2, 199 (47.6%) had brain metastasis, including 148/199 with symptomatic brain metastasis (74.4%). The majority of patients (70.6%) had a good performance status (i.e., ECOG-PS 0 or 1) (Table 1). Focusing on CNS metastasis (i.e., brain metastasis plus leptomeningeal metastasis), patients with CNS lesions had a median age at the index date of 67.0 years (range: 28–89) versus 72.0 years (range: 36–93) for patients without CNS metastasis. Moreover, 20.5% of patients with CNS metastasis were over 75 years old versus 35.5% for those without CNS metastasis (Table 2). Regarding cerebral irradiation, 3/198 patients (1.51%) were treated with whole-brain radiation therapy (WBRT) before osimertinib in L1 versus 20/426 patients (4.7%) in L2.

*EGFR* mutations were distributed as follows: *EGFR-L858R* = 260/624 patients (41.7%); *EGFR-DEL19 =* 364/624 patients (58.3%). The gatekeeper *T790M* mutation in exon 20 was identified in 257/426 patients (60.3%) receiving osimertinib in L2.

### 3.2. Progression-Free Survival and Overall Survival According to the Line of Osimertinib Adminsitration

Median PFS according to the line of osimertinib initiation was 12.4 months (95% CI [10.7–14.7 months]) for L1 and 7.4 months (95% CI [6.2–8.7 months]) for L2. Regarding the prognostic impact of CNS metastases in patients treated with osimertinib in L1, unadjusted median PFS was 11.3 months (95% CI [10.1–15.7 months]) for patients with cerebral metastases and 12.7 months (95% CI [9.0–16.9 months]) for patients without cerebral metastases (HR = 0.88; 95% CI [0.63–1.22]; *p* = 0.4). For patients receiving osimertinib in L2, median PFS was 6.3 months (95% CI [5.4–7.9 months]) and 8.7 months (95% CI [6.8–9.9 months]) (HR = 0.89; 95% CI [0.73–1.09]; *p* = 0.3) for patients with and without cerebral metastases, respectively (Figure 2). Median OS from the advanced diagnosis date was 28.5 months (95% CI [26.3–38.7 months]) for osimertinib in L1 and 29.9 months (95% CI [28.6–31.8 months]) for patients treated with first- or second-generation TKIs in L1 (n = 1262) (HR = 0.93; 95% CI [0.75–1.16]; *p* = 0.50). In patients treated with osimertinib in L1, unadjusted median OS was 27.1 months (95% CI [22.0–30.2 months]) for patients with cerebral metastases and 38.7 months (95% CI [26.3–52.8 months]) for patients without cerebral metastases (HR = 0.73; 95% CI [0.48–1.11]; *p* = 0.15) (Figure 3). Of the 327 patients without CNS metastases at the index date (i.e., the date of osimertinib initiation), the estimated probability of being cerebral metastasis-free at 18 months was 86.9% (95% CI [79.4–94.4%]) for osimertinib in L1 and 75.7% (95% CI [68.4–83.0%]) for osimertinib in L2. Median time to CNS progression was not reached (NR) (95% CI [NR–NR]) for osimertinib in L1 and 45.2 months (95% CI [34.6–NR]) for osimertinib in L2 (HR = 1.66; 95% CI [0.94–2.92]; Figure 4).

### 3.3. Subsequent Treatment Received after Osimertinib

Of note, 356/624 patients (57.1%) did not receive a subsequent treatment line after osimertinib administration. When osimertinib was administered in L1, 131/198 (66.2%) did not receive a subsequent treatment line, which was the case for 225/426 (52.8%) when osimertinib was given in L2. For patients receiving subsequent treatment after osimertinib, 75.4% received a platinum-based chemotherapy regimen; 19.1% received an *EGFR*-TKI or *MET*-TKI alone or in combination; 23.8% received a combination containing a vascular endothelial growth factor (VEGF) inhibitor; 9.3% received a regimen including an immune checkpoint inhibitor; and 4.5% a non-platinum-based chemotherapy.

### 3.4. Prognostic Factors Associated with Overall Survival

Focusing on the prognostic factors associated with OS(id) in independent multivariable analysis according to the line of osimertinib administration (L1 and L2), a good ECOG status (i.e., 0–1 versus ≥2) in L1 (*p* = 0.016) and L2 (*p* < 0.001) and an exon 19 deletion (versus L858R mutation in exon 21) in L1 (*p* = 0.019) and L2 (*p* < 0.001) were identified as good prognostic factors. Additionally, the female gender was associated with better prognosis for osimertinib administered in L1 (*p* = 0.03); the presence of liver metastasis (*p* = 0.004) and PD-L1 positive testing (*p* = 0.018) were associated with worse prognosis for osimertinib administered in L2 (Appendix A).

## 4. Discussion

This real-world multicenter retrospective study focused on the treatment sequences and outcomes of 624 patients treated for *EGFR*-mutated advanced NSCLC. The median OS was 28.5 months (95% CI [26.3–38.7 months]) for 198 patients treated with osimertinib monotherapy in the first line (L1) and 29.9 months (95% CI [28.6–31.8 months]) for 1262 patients treated with first- or second-generation TKIs in L1, among whom 426 patients received osimertinib in the second line (L2) (HR = 0.93; 95% CI [0.75–1.16]; *p* = 0.50). The median OS of patients receiving osimertinib in L1 in our real-world cohort was shorter than in the FLAURA trial (38.6 months (95% CI [34.5–41.8 months] [7]. This discrepancy is probably linked to different patient baseline characteristics: in the FLAURA trial, patients were younger (median age= 64 years (26–85) vs. 70.0 (28–93)), had better general status (ECOG PS 2 = 0% vs. ECOG PS 2 = 29.4%) than in our real-world cohort. More importantly, 52/198 (26.3%) patients treated with osimertinib in L1 in this real-world cohort had symptomatic brain metastases, whereas these patients were excluded from the FLAURA study. Recent data confirms this assessment. Indeed, from a cohort of 311 patients treated with L1 osimertinib in Canada, Connor Wells et al. classified patients according to whether or not they met the inclusion criteria of the FLAURA study. The median OS for the entire cohort was 27.4 months (95% CI [23.8–30.1]). The median OS for the 137 (44%) patients deemed ineligible for the FLAURA trial was 18.4 months shorter than that of eligible patients (15.8 vs. 34.2 months; *p* < 0.001) [14].

Moreover, OS from the APPLE trial of osimertinib versus gefitinib followed by osimertinib in advanced *EGFR*-mutant NSCLC is almost the same in both strategies (HR = 1.01; 90% CI [0.61–1.68]), with a 18-month survival probability of 84% and 82.3%, respectively [15]. One main explanation for the discrepancies observed between the control arm of FLAURA and that of the APPLE trial is the proportion of patients treated with osimertinib in L2 after disease progression under first- or second-generation TKIs. In the FLAURA trial, 47% of patients benefiting from the first subsequent therapy were treated with osimertinib after progression with gefitinib or erlotinib. This rate is much lower than in the APPLE trial (73%), which is in line with what is expected in the real-world setting, where patients may be monitored more closely, leading to a higher rate of rebiopsy and detection of the *EGFR T790M* mutation in patients receiving first- or second-generation TKIs, which appears to be present in up to 50–75% of cases [16,17,18]. Therefore, the control arm of the FLAURA trial may underestimate the benefit of an appropriate TKI sequencing approach in this population. Of note, in our real-world cohort, 66.2% of the patients receiving osimertinib in L1 and 29.9% of patients treated with first- or second-generation TKIs in L1 did not receive subsequent therapy after progression. These results are in line with those of the FLAURA trial, where 146/279 patients (52%) did not receive subsequent therapy after frontline osimertinib versus 97/277 patients (35%) treated with gefitinib or erlotinib. Otherwise, osimertinib presents many advantages compared to first- and second-generation *EGFR*-TKIs. First of all, osimertinib has higher CNS penetration [19]. Preventing or delaying the onset of CNS metastasis is particularly important as it clearly impairs the patients’ quality of life and cognition [20]. Moreover, the safety profile of osimertinib is better than that of gefitinib or erlotinib. In the FLAURA trial, adverse events of grade 3 or higher were less frequent with osimertinib than with other *EGFR*-TKIs (34% vs. 45%) [8].

The next step for the treatment of *EGFR*-mutated patients will be the implementation of chemotherapy in combination with osimertinib in the first metastatic line, according to the results of the FLAURA-2 trial [21]. Indeed, the combination of platinum-pemetrexed chemotherapy plus osimertinib significantly extended PFS as assessed by the investigators, by 8.8 months compared with osimertinib monotherapy (HR = 0.62; 95% CI [0.49–0.79]; *p* < 0.001) [21]. The addition of chemotherapy in combination with osimertinib seems particularly valuable for patients with CNS metastases. This combination delays intracranial disease progression, irrespective of baseline CNS metastasis status. In the full CNS analysis set of the FLAURA-2 trial, the HR for CNS progression or death was 0.58, 95% CI [0.33–1.01], in patients with baseline CNS metastases and 0.67, 95% CI [0.43–1.04], in patients without baseline CNS metastases, both favoring chemo-osimertinib treatment. The CNS objective response rates were 73% (64 to 81%) in the combination arm versus 69% (59 to 78%) in the osimertinib monotherapy arm; 59% versus 43% had CNS complete response [22]. Some other considerations could be of interest as co-mutation profile (*TP53* mutant vs. *TP53* wild-type) to propose therapeutic intensification to patients.

Our study has some limitations. (1) According to the inclusion period, most patient were treated with osimertinib in L2; (2) due to the low search rate for associated mutations, we could not evaluate their prognostic and predictive impact, particularly for *TP53* status; (3) the ESME database is not appropriate to evaluate the resistance mechanisms induced by osimertinib. Currently, the mechanisms of resistance to osimertinib are mainly the occurrence of MET amplification (15–20%), the occurrence of a new mutation in the EGFR gene (10%), the acquisition of resistance mutations in the MAPKinase and PIK3 signaling pathways (10–15%), and a possible histological switch (10–15%) of cases towards small-cell lung cancer or a squamous NSCLC profile [23,24]. There are also other resistance mechanisms that are being extensively studied in basic and translational research. These include drug-tolerant persister cells, which can survive under therapeutic pressure by having a slow proliferative rate with an altered cellular metabolism [25,26]. These cell subtypes may serve as a reservoir from which various resistance mechanisms can emerge [27]. Moreover, growing evidence suggests that the tumor microenvironment may play a role in the acquired resistance to osimertinib. Specifically, an immunosuppressive environment may contribute to resistance against third-generation EGFR tyrosine kinase inhibitors (TKIs) [28].

## 5. Conclusions

This large real-world cohort indicates that advanced *EGFR*-mutant NSCLC patients have a poorer prognosis than patients included in clinical trials. Median PFS was 12.4 months (95% CI [10.7–14.7]) for osimertinib administred in L1 in our cohort. It also confirms the poor prognosis of patients with cerebral metastases. The optimization of treatment strategies for these patients remains a major challenge in *EGFR*-mutated NSCLC.

## Figures and Tables

**Figure 1 cancers-16-03563-f001:**
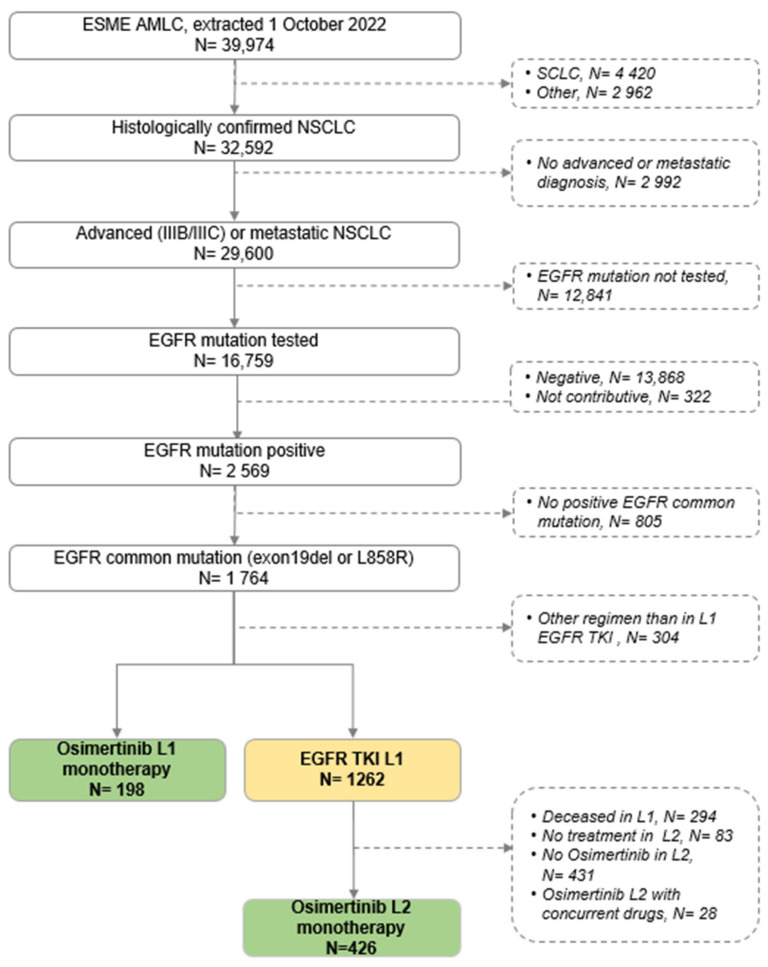
Flow-chart. AMLC = advanced or metastatic lung cancer; SCLC = small-cell lung cancer; TKI = tyrosine kinase inhibitor.

**Figure 2 cancers-16-03563-f002:**
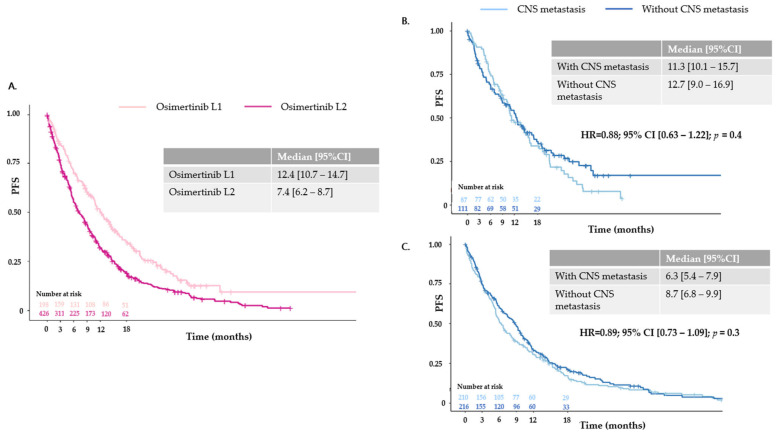
Progression-free survival: (**A**) according to osimertinib administration line; (**B**) according to central nervous system metastasis status at index date (i.e., start date of osimertinib) in L1; (**C**) according to central nervous system metastasis status at index (i.e., start date of osimertinib) date in L2.

**Figure 3 cancers-16-03563-f003:**
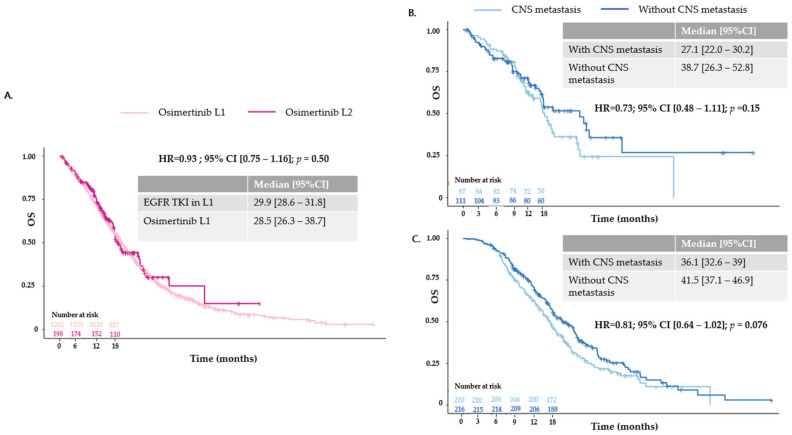
Overall survival: (**A**) OS: from the locally advanced or metastatic diagnosis date, whether patients were treated with osimertinib or first- or second-generation TKIs in L1. (**B**) OSid: according to central nervous system metastasis status at index date (i.e., start date of osimertinib) in L1. (**C**) OSid: according to central nervous system metastasis status at index (i.e., start date of osimertinib) date in L2.

**Figure 4 cancers-16-03563-f004:**
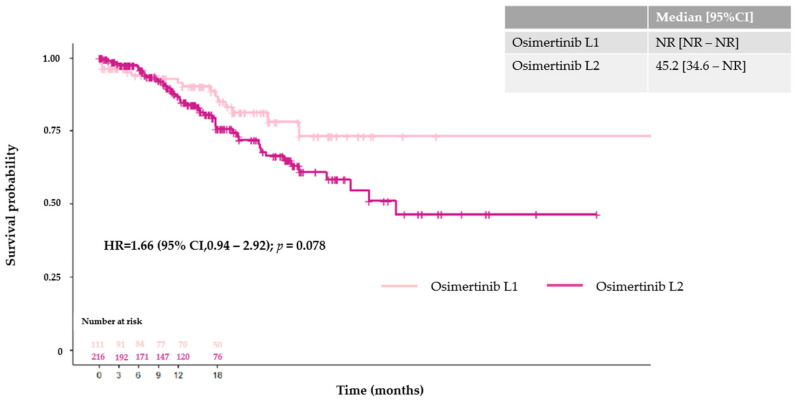
Time to central nervous system metastasis for patients treated with osimertinib.

**Table 1 cancers-16-03563-t001:** Patient characteristics according to line of initiation of osimertinib.

	Study Population Overall (n = 624)	Osimertinib in L1 (n = 198)	Osimertinib in L2 (n = 426)
**Age at index date (yr)**			
Median (range)	70.0 (28.0; 93.0)	71.0 (34.0; 93.0)	69.0 (28.0; 92.0)
>75 yr—no (%)	177 (28.4)	58 (29.3)	119 (27.9)
**Female sex—no. (%)**	458 (73.4)	151 (76.3)	307 (72.2)
**Smoking status—no. (%); (n = 577)**			
Current smoker	48 (8.3)	11 (6.2)	37 (9.3)
Former smoker	189 (32.8)	63 (35.4)	126 (31.6)
Never	340 (58.9)	104 (58.4)	236 (59.1)
**Number of pack-years—Median (Q1–Q3)**			
Current smoker	24.5 (15.0; 30.0)	30.0 (20.0; 40.0)	22.0 (13.0; 30.0)
Former smoker	15.0 (6.0; 30.0)	15.0 (6.0; 30.0)	15.0 (6.0; 25.0)
**ECOG performance status score—** **no. (%); (n = 289)**			
0–1	204 (70.6)	75 (65.2)	129 (74.1)
≥2	85 (29.4)	40 (34.8)	45 (25.9)
**Histologic type—no (%)**			
Adenocarcinoma	589 (94.4)	187 (94.4)	402 (94.4)
Squamous carcinoma	15 (2.4)	3 (1.5)	12 (2.8)
Other °	20 (3.2)	8 (4.0)	12 (2.8)
**PD-L1 tumor proportion score—no (%); (n = 416)**			
<1%	190 (45.7)	72 (45.3)	118 (45.9)
1–49%	140 (33.6)	61 (38.4)	79 (30.7)
≥50%	74 (17.8)	22 (13.8)	52 (20.3)
Not contributive	12 (2.9)	4 (2.5)	8 (3.1)
***EGFR* mutation type—no (%)**			
Exon 19 deletion	364 (58.3)	104 (52.5)	260 (61.0)
*L858R*	260 (41.7)	94 (47.5)	166 (39.0)
**Metastases status—no (%)**			
Brain (asymptomatic/symptomatic)	282 (46.7): 82/200	83 (44.6): 31/52	199 (47.6): 51/148
Meninges	32 (5.3)	6 (3.2)	26 (6.2)
Contralateral lung	217 (35.9)	57 (30.6)	160 (38.3)
Bone	372 (61.6)	106 (57.0)	266 (63.6)
Liver	143 (23.7)	33 (17.7)	110 (26.3)
Pleura	114 (18.9)	32 (17.2)	82 (19.6)
Adrenal gland	81 (13.4)	29 (15.6)	52 (12.4)
Other	166 (27.5)	42 (22.6)	124 (29.7)
**EGFR-TKI previously administered—no (%)**			
Gefitinib	145 (23.2)	0	145 (34.0)
Erlotinib	168 (26.9)	0	168 (39.4)
Afatinib	71 (11.4)	0	71 (16.7)

° Undifferentiated carcinoma; neuroendocrine large cell carcinoma; others.

**Table 2 cancers-16-03563-t002:** Patient characteristics according to CNS metastasis status.

	Patient Without CNS Metastasis (n = 327)	Patient With CNS Metastasis (n = 297)
**Age at index date (yr)**		
Median (range)	72.0 (36.0; 93.0)	67.0 (28.0; 89.0)
>75 yr—no (%)	116 (35.5)	61 (20.5)
**Female sex—no. (%)**	234 (71.6)	224 (75.4)
**Smoking status—no. (%); (n = 577)**		
Current smoker	24 (7.9)	24 (8.8)
Former smoker	95 (31.3)	94 (34.4)
Never	185 (60.9)	155 (56.8)
**Number of pack-years—Median (Q1–Q3)**		
Current smoker	30.0 (20.0; 35.0)	20.0 (15.0; 30.0)
Former smoker	15.5 (7.0; 30.0)	10.0 (6.0; 20.0)
**ECOG performance-status score—no. (%);**	n = 55	n = 53
0–1	43 (78.2)	43 (81.1)
≥2	12 (21.8)	10 (18.9)
**Histologic type—no (%)**		
Adenocarcinoma	306 (93.6)	283 (95.3)
Squamous carcinoma	11 (3.4)	4 (1.3)
Other °	10 (3.1)	10 (3.4)
**PD-L1 tumor proportion score—no (%); (n = 416)**		
<1%	190 (45.7)	72 (45.3)
1–49%	140 (33.6)	61 (38.4)
≥50%	74 (17.8)	22 (13.8)
Not contributive	12 (2.9)	4 (2.5)
***EGFR* mutation type—no (%)**		
Exon 19 deletion	186 (56.9)	178 (59.9)
*L858R*	141 (43.1)	119 (40.1)
**EGFR-TKI previously administered—no (%) ***	n = 216	n = 210
Gefitinib	81 (37.5)	87 (41.4)
Erlotinib	30 (13.9)	54 (25.7)
Afatinib	105 (48.6)	79 (37.6)

° Undifferentiated carcinoma; neuroendocrine large cell carcinoma; others. * Only for patients receiving osimertinib in L2.

## Data Availability

The datasets analyzed in the current study were extracted from the ESME LC Data Platform. In accordance with the ethical and legal requirements related to the ESME data warehouse, individual data from the ESME databases cannot be made available. For any specific request, please contact the corresponding author. Each request will be examined on a case-by-case basis by the scientific committee. During the preparation of this work, the author(s) used open AI Chat GPT in order to improve language and readability. After using this tool/service, the author(s) reviewed and edited the content as needed and take(s) full responsibility for the content of the publication.

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
