# Peer review of "Optimizing Treatment Strategies for Egfr-Mutated Non-Small-Cell Lung Cancer Treated with Osimertinib: Real-World Outcomes and Insights"

_cancers, 2024, doi:10.3390/cancers16213563_

Round 1
Reviewer 1 Report
Comments and Suggestions for Authors
This retrospective study analyzed data from 624 patients with EGFR-mutant advanced NSCLC treated with osimertinib using the ESME platform. Patients received osimertinib either as first-line (L1) or second-line (L2) therapy after prior treatment with first- or second-generation TKIs. The median progression-free survival (PFS) was 12.4 months for L1 and 7.4 months for L2, with overall survival (OS) of 28.5 months for L1 and 29.9 months for L2. Patients with brain metastases had worse outcomes, and overall survival in real-world data was poorer than observed in the FLAURA trial. This is an important article for the field of EGFR-mutant lung cancer. However, several comments still need to be addressed.
Minor Comments:
-
The authors should address the mechanisms of resistance to osimertinib, such as the involvement of APOBECs, AXL, and low-fidelity polymerases.
-
The role of Drug-Tolerant Persisters (DTPs), including both cycling and non-cycling persisters, in regulating osimertinib resistance needs to be discussed.
-
The impact of osimertinib on the tumor microenvironment and its potential contribution to resistance should also be explored.
Author Response
We would like to thank reviewer 1 for the time he devoted to proofreading and providing valuable comments on our article. We share his enthusiasm regarding the fact that this is an important article for the field of EGFR-mutant lung cancer. Here is a point by point response to each comments:
Minor Comments:
- The authors should address the mechanisms of resistance to osimertinib, such as the involvement of APOBECs, AXL, and low-fidelity polymerases.
We fully agree with the reviewer 1 that the involvement of APOBECs, AXL, and low-fidelity polymerases are important to study in understanding the mechanisms of osimertinib resistance. However, our paper remains extremely focused on routine therapeutic strategies and the results of therapeutic sequences. We feel that tackling this subject is too far removed from the study's objective.
- The role of Drug-Tolerant Persisters (DTPs), including both cycling and non-cycling persisters, in regulating osimertinib resistance needs to be discussed
Indeed, the role of DTPs is fundamental to understanding the mechanisms of resistance to TKIs in EGFR-mutated models. We have added a summary paragraph on this subject among the mechanisms of resistance to osimertinib. Page 11; lines 316-324
“ Currently, the mechanisms of resistance to osimertinib are mainly the occurrence of MET amplification (15-20%), the occurrence of a new mutation in the EGFR gene (10%), the acquisition of resistance mutations in the MAPKinase and PIK3 signaling pathways (10-15%), and a possible histological switch (10-15%) of cases towards a small-cell lung cancer or squamous-NSCLC profile (21-22). There are also other resistance mechanisms that are being extensively studied in basic and translational research. These include drug-tolerant persister cells, which can survive under therapeutic pressure by having a slow proliferative rate with altered cellular metabolism (23-24). These cell subtypes may serve as a reservoir from which various resistance mechanisms can emerge (25).”
- The impact of osimertinib on the tumor microenvironment and its potential contribution to resistance should also be explored.
Immunosuppressive microenvironment is indeed a potential marker that contribute to osimertinib resistance. A sentence has been added in this direction: Page 11; lines 324-328
“Moreover, growing evidence suggests that the tumor microenvironment may play a role in the acquired resistance to osimertinib. Specifically, an immunosuppressive environment may contribute to resistance against third-generation EGFR tyrosine kinase inhibitors (TKIs)(26).”

Reviewer 2 Report
Comments and Suggestions for Authors
The manuscript titled “Optimizing treatment strategies for EGFR-mutated NSCLC treated with osimertinib: real-world outcomes and insights” detailed a large cohort of consecutive patients with advanced, metastatic EGFR-mutant NSCLC to report therapeutic strategies and outcomes of patients treated in a real-world setting, with a focus on CNS metastases. Authors indicated that patients with advanced EGFR-mutant NSCLC have a poorer prognosis than those included in clinical trials, while also confirming the poor prognosis of patients with cerebral metastases. Manuscript is interesting.
While, there are point that need to be addressed as following:
1. Please pay significant attention to abstract to specify what authors are trying to achieve.
2. Please pay significant attention to the subheads as they sound very generalized. Eg. “Patient Population”, “Study Population and Patient Characteristics”, etc.
3. Please provide details/guidelines for patient ethical consent followed in the study.
4. Please provide details of Conflicts of Interest.
5. The manuscript requires a significant attention specifically in result section to improve punctuations, grammar and the readability.
Comments on the Quality of English LanguagePlease see comment 5 above.
Author Response
We would like to thank reviewer 2 for the time he devoted to proofreading and providing valuable comments on our article. Here is a point by point response to each comments:
- Please pay significant attention to abstract to specify what authors are trying to achieve.
The aim of our abstract was to focus on the poorer real-wolrd survival data than in clinical trials for EGFR-mutant patients, and on the poor prognostic factor represented by the presence of brain metastases. If reviewer 2 considers that other data should be highlighted, we are inclined to reconsidering our abstract.
- Please pay significant attention to the subheads as they sound very generalized. Eg. “Patient Population”, “Study Population and Patient Characteristics”, etc.
We agree with reviewer 2 that some of our subheads sound very generalized. We have modified several subheads as follows:
3.1. Characteristics of EGFR-mutated patients treated with osimertinib for metastatic NSCLC
3.2. Progression-free survival and overall survival according to the line of osimertinib adminsitration
3.3. Subsequent Treatment received after osimertinib
3.4. Prognostic factors associated with overall survival
- Please provide details/guidelines for patient ethical consent followed in the study.
we apologize for not having specified details for patient ethical consent in our manuscript. Here is the institutional review board statement and the informed consent statement: page 12; lines 348-355
“Institutional Review Board Statement: Unicancer manages the ESME AMLC data platform in accordance with current best practice guidelines. In compliance with French regulations, the ESME Data platform was authorized by the French data protection authority (initial authorization no. DE-2017-397 and subsequent amendment dated 14 October 2019, in accordance with GDPR).
Informed Consent Statement: In accordance with the authorizations issued by the French authorities for the ESME AMLC dataplatform and in compliance with the regulation in force for such retrospective registry, no formal dedicated informed consent is required, but all patients have approved the analysis of their electronically recorded data.”
- Please provide details of Conflicts of Interest.
The section of conflict of interest have been completed pages 12-13; lines 381-408
- The manuscript requires a significant attention specifically in result section to improve punctuations, grammar and the readability.
Some modification have been made to improve the readability of the manuscript

Reviewer 3 Report
Comments and Suggestions for Authors
Dear Authors,
I really liked your manuscript and have just some small remarks:
1)M+M 2.1 please separate the inclusion criteria point by point. Also exclusion criteria should be included here;
2) Discussion. Please, remove reference to the Fig 1 here, this is not a Result part. And, - underline at the end the novelty and the significance of this done research!
3) I didnt like a bit this small number of references, but ok, - otherwise the manuscript is well elaborated, concentrated and looks good...
Author Response
We would like to thank reviewer 3 for the time he devoted to proofreading and providing valuable comments on our article. We also appreciate that the reviewer liked our manuscript. Here is a point by point response to each comments:
1)M+M 2.1 please separate the inclusion criteria point by point. Also exclusion criteria should be included here;
Thanks to the reviewer comment, we modified our M&M 2.1 section as follows: page 3 lines 99-113
“In this study, patient selection focused on patients with the following inclusion criteria:
- Histologically-confirmed advanced NSCLC,
- Common EGFR activating mutations (i.e., exon 19 deletion or L858R mutation in exon 21),
- Patients treated with osimertinib in the first (L1) or second (L2; i.e., administered after first- or second-generation TKIs) line of treatment for advanced disease.
- Patients were either newly diagnosed or relapsed from early-stage disease.
Patients with one of the following exlusion criteria were excluded of the study:
- Uncommon EGFR activating mutations identified
- Patients treated with another regimen than EGFR TKIs in first metastatic line
- Patients who received first- or second-generation EGFR TKIs as first-line treatment for metastatic disease were excluded if they had not been treated with osimertinib monotherapy as second-line therapy.
2) Discussion. Please, remove reference to the Fig 1 here, this is not a Result part. And, - underline at the end the novelty and the significance of this done research!
Indeed, results should not appear in the discussion section. Figure 1 reference has been removed.
3) I didnt like a bit this small number of references, but ok, - otherwise the manuscript is well elaborated, concentrated and looks good...
We would like to thanks the reviewer for this comment. Eight references have been added to the manuscript during the revision (references 10-11 and 23-28)

Reviewer 4 Report
Comments and Suggestions for Authors
Dear respected Editor,
Thanks for this invitation to review the manuscript “Optimizing Treatment Strategies for Egfr-Mutated Nsclc Treated with Osimertinib: Real-World Outcomes and Insights”. This paper includes a good survey. The authors reported a review study of 624 EGFR-Mutated NSCLC patients who were treated with osimertinib drug. After reviewing this manuscript, it needs some modifications to become fit for publication.
The manuscript will be reviewed again after major revision. Specific comments are listed below:
1- I recommend changing the title to:
“Optimizing Treatment Strategies for Osimertinib-treated EGFR-Mutated Non-Small Cell Lung Cancer
Patients: Real-World Outcomes and Insights”
2-Please correct this header “Cancers 2022, 14, x. “ to “Cancers 2024, 14, x.
3-Please to improve the strength of this paper, add an illustrated graph that includes the number of NSCLC patients compared with EGFR-Mutated NSCLC patients and Osimertinib-treated EGFR-Mutated NSCLC patients.
4-The quality of the right side of Fig.1 needs to be improved, change its font style
5-Please revise the significance of numbers in Table 1 and Table 2
6-The quality of Fig.2-Fig.4 is very poor
7-Conclusion needs to be rewritten concisely and quantitively.
8- Please revise the English writing throughout the whole manuscript. Please double-check on grammar, missing articles, punctuation, and so on.
Comments on the Quality of English Language
Please revise the English writing throughout the whole manuscript. Please double-check on grammar, missing articles, punctuation, and so on.
Round 2
Reviewer 2 Report
Comments and Suggestions for Authors
In the updated manuscript, Optimizing Treatment Strategies for EGFR-Mutated NSCLC Treated with Osimertinib: Real-World Outcomes and Insights, the authors have thoroughly addressed all previous concerns. The revised manuscript is now convincing, presenting a comprehensive cohort study of advanced EGFR-mutant NSCLC, which highlights significantly worse outcomes, especially in patients with CNS metastases, compared to clinical trials.
Author Response
“In the updated manuscript, Optimizing Treatment Strategies for EGFR-Mutated NSCLC Treated with Osimertinib: Real-World Outcomes and Insights, the authors have thoroughly addressed all previous concerns. The revised manuscript is now convincing, presenting a comprehensive cohort study of advanced EGFR-mutant NSCLC, which highlights significantly worse outcomes, especially in patients with CNS metastases, compared to clinical trials.”
We would like to thank reviewer 2 for the time he devoted to proofreading and providing valuable comments on our article. We are happy to see that the first round of reviewing has been considered suitable for publication by the reviewer 2.

Reviewer 4 Report
Comments and Suggestions for Authors
5-Please revise the significance of numbers in Table 1 and Table 2
We're really sorry but we don't understand the reviewer's comment. Is it possible to clarify what the reviewer wants us to change by the term “significance of numbers”?
I mean that all numbers have the same digits
For example, 10.01 (4 digits ), all numbers in the table have the same digits
20.1 should be written 20.10
I hope my suggested numbers are clear for authors
